# SPOT THE CRITICAL WORDS:
# TEXT-GUIDED VISUAL TOKEN PRUNING FOR EFFICIENT LARGE VISION-LANGUAGE MODEL INFERENCE

## ABSTRACT

The computational efficiency of Large Vision-Language Models (LVLMs) is severely hampered by the processing overhead of massive visual tokens. While token pruning emerges as a promising solution, prevailing methods that rely on text-visual cross-attention suffer from attention shift, a phenomenon where attention maps fail to accurately localize instruction-relevant regions, retaining significant visual redundancy. To address this issue, we propose **TextScythe**, a intuitive yet potent pruning framework that first identifies *vision-critical text tokens* through an entropy-based analysis of cross-modal cosine similarity, effectively distilling user's instructions. It then selects visual tokens exhibiting outlier-level similarity to these critical text tokens. To preserve contextual completeness, a diversity-aware mechanism supplements background tokens based on their intrinsic attention scores. Extensive experiments show that TextScythe achieves state-of-the-art performance across various benchmarks, enabling an extreme 88.9% token reduction in LLaVA while retaining 96.6% of the original accuracy, thereby establishing an efficient and effective deployment paradigm for LVLMs. *The code will be released.*

## 1 INTRODUCTION

Recent advances in Large Vision-Language Models (LVLMs) have achieved remarkable success in various vision-language tasks (Liu et al., 2024c; Wang et al., 2024b;c). However, these models can be highly computationally intensive, limiting their practicality in resource-constrained environments (Liu et al., 2024a). One important reason for such huge costs is that these models typically handle a large number of visual tokens representing input images, especially high-resolution images (Li et al., 2024c) and long videos (Lin et al., 2023), but a significant portion of these visual tokens are redundant or irrelevant to a specific user's instructions, presenting a substantial opportunity for compression.

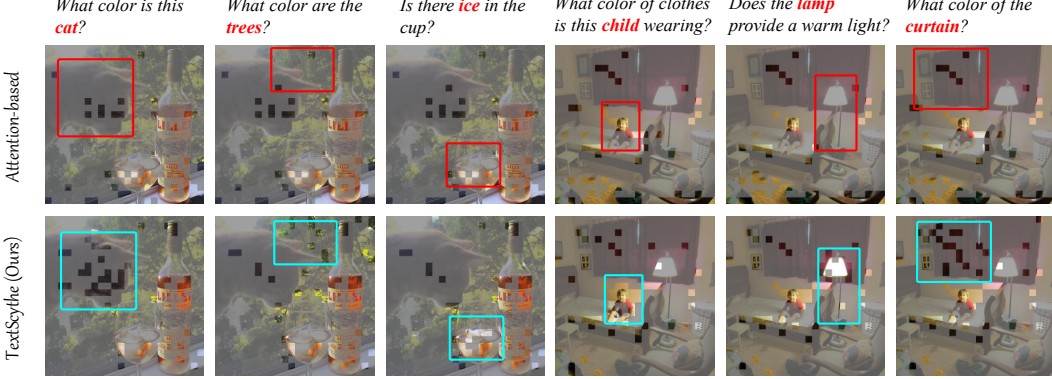

Figure 1: Comparison of different pruning methods. Attention-based methods tend to suffer from text-visual attention shift, leading to inaccurate focus on query-relevant regions. In contrast, our proposed TextScythe accurately preserves more detailed visual tokens relevant to user's instructions.

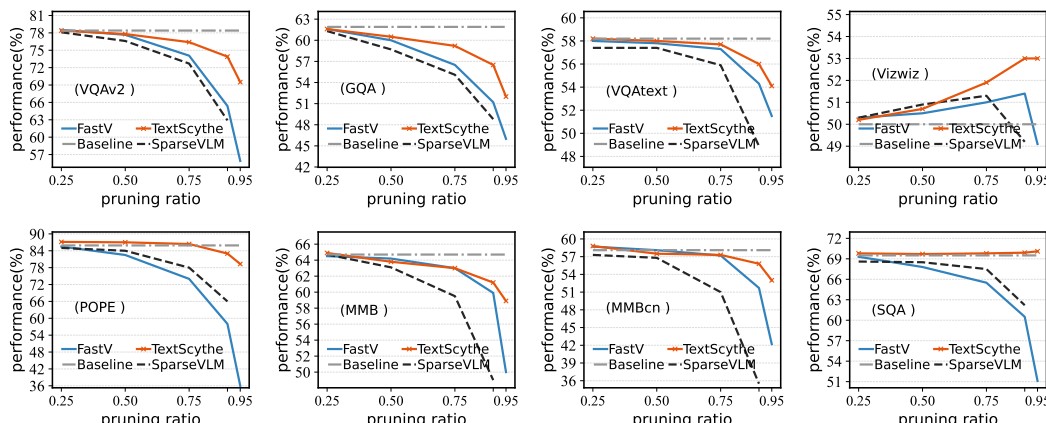

Figure 2: Relationship between performance and pruning ratios of different baseline methods. As the token pruning ratio grows, the performance of these attention-based strategies degrades dramatically, while TextScythe maintains the favorable performance even at 90% and 95% of pruning ratios.

Significant efforts have been devoted to reducing LVLM inference costs through visual token pruning, with existing methods classified into two paradigms. The first identifies high-attention visual tokens as critical while discarding low-scoring ones (Chen et al., 2024; Xing et al., 2025; Zhang et al., 2024c). The second eliminates redundancy based on inter-token feature similarity (Wen et al., 2025b; Alvar et al., 2025; Jeddi et al., 2025). However, both approaches exhibit limitations: similarity-based methods completely ignore cross-modal interactions with text tokens, failing to achieve instruction-aware dynamic pruning. Although attention-based methods capture text-visual interactions, they suffer from cross-attention shift, which prevents accurate identification of instruction-relevant visual tokens (as shown in Figure 1), leading to suboptimal post-pruning performance.

To address these challenges, we introduce TextScythe, a plug-and-play vision token pruning framework that enables efficient multi-modal reasoning by jointly optimizing instruction relevance and feature diversity. As illustrated in Figure 2, the proposed TextScythe maintains robust performance even under high pruning ratios, significantly outperforming existing methods. Specifically, our approach first identifies key text tokens relevant to the image based on the entropy of the cosine similarity between text and visual tokens. Visual tokens strongly correlated with these key text tokens are prioritized for retention. Then, to ensure that background information is not overlooked, we select highly [CLS] attention and unique tokens from the remaining visual tokens as a complement, thus maximizing the retention of effective information from the image. Unlike text-visual attention-based methods that prune within LLMs, our approach leverages visual cues to prune before the language model, ensuring compatibility with various attention optimization techniques such as FlashAttention (Dao et al., 2022), thus achieving higher inference efficiency.

In summary, the contributions of our work can be summarized as follows:

- We identify the *text-visual attention shift* problem in prior pruning methods and propose *semantic cosine similarity* as a more robust signal for quantifying cross-modal relevance.
- We introduce an *entropy-ratio* metric and an *instruction-length* aware adaptive thresholding mechanism. These components collaboratively enable the precise distillation of instructions into a minimal subset of vision-critical text tokens, serving as faithful anchor points for token pruning.
- We propose TextScythe that integrates instruction-aware pruning with a *diversity-aware supplementation* of background tokens. Comprehensive experiments demonstrate that TextScythe outperforms SOTA methods and achieves superior performance-efficiency trade-offs across various benchmarks.

## 2 RELATED WORK

### 2.1 MULTIMODAL LARGE LANGUAGE MODELS

Large Language Models (LLMs) (Bai et al., 2023; Jiang et al., 2023; Ouyang et al., 2022; Touvron et al., 2023) have recently achieved remarkable success, leading to a growing trend of extending

their powerful reasoning capabilities to multimodal understanding tasks, ultimately giving rise to Multimodal Large Language Models (MLLMs) (Liu et al., 2024c; Li et al., 2024a; Wang et al., 2024a; Bai et al., 2025b; Chen et al., 2025; Zhu et al., 2025). These models typically encode visual inputs into tokens to fully leverage LLMs' capabilities. While enabling visual perception, this approach introduces substantial computational overhead from long visual token sequences. For example, LLaVA-1.5 (Liu et al., 2024a) converts a $336 \times 336$ image into 576 tokens, while its high-resolution variant LLaVA-NeXT (Liu et al., 2024b) generates 2,880 tokens from double-resolution images. In video understanding scenarios, models like LongVA (Zhang et al., 2024a) can produce ultra-long sequences exceeding 200K visual tokens. Thus, it is crucial to accelerate MLLM inference.

## 2.2 VISUAL TOKEN COMPRESSION

One effective approach to optimizing MLLM inference involves reducing the predominantly visual tokens in input sequences. Compared to text dense with information, visual signals exhibit greater spatial redundancy (Marr, 2010). While some works attempt visual token compression through vision-text prefusion (Li et al., 2024b; Hu et al., 2024; Cai et al., 2024; Zhang et al., 2025), these methods require architectural modifications and additional training, thereby increasing computational costs. Alternative training-free approaches, known as token pruning, remove redundant visual tokens during inference. FastV (Chen et al., 2024) first identified the redundancy in LVLMs and proposed pruning low-attention visual tokens after the second layer of the language model. SparseVLM (Zhang et al., 2024c) eliminates text prompt interference and employs more accurate text attention for progressive visual token sparsification. However, such text-visual attention-based methods suffer from attention shift issues (Zhang et al., 2024b; Wen et al., 2025a) that compromise pruning accuracy, and they remain incompatible with efficient attention implementations like FlashAttention (Dao et al., 2022; Dao, 2023). Other studies (Wen et al., 2025b; Alvar et al., 2025; Jeddi et al., 2025) prune tokens based on inter-token feature similarity, but ignore the critical relevance between visual tokens and user instructions, leading to suboptimal performance. However, our proposed TextScythe addresses these limitations by simultaneously optimizing instruction relevance and token distinctiveness for more effective visual pruning while maintaining hardware acceleration compatibility.

## 3 METHODOLOGY

### 3.1 PRELIMINARY

#### 3.1.1 ARCHITECTURE OF VLMs.

VLMs typically comprise three core components: a visual encoder, a modality projector, and a language model. The visual encoder (*e.g.*, a pretrained ViT) transforms input images into visual tokens, which are then aligned with text tokens via the modality projector and fed into the LLM to generate responses by integrating visual and textual information. The computational complexity of VLMs, formulated as FLOPs $= T \times (4nd^2 + 2n^2d + 2ndm)$ where $T$ is the number of transformer layers, $n$ is the sequence length, $d$ is the hidden dimension size, and $m$ is the intermediate size of FFN, highlights the quadratic dependence on $n$. In typical VLM tasks, $n$ covers prompt, text, and visual tokens. With visual tokens dominating sequence length, reducing them is critical for efficiency.

#### 3.1.2 [CLS] ATTENTION IN VISUAL ENCODER

Visual encoders, such as CLIP (Radford et al., 2021), often employ a global attention mechanism to capture relationships between image patches. Given a sequence of image patch embeddings $\mathbf{X} = [\mathbf{x}_{\text{cls}}; \mathbf{x}_{\text{img}}^1; \mathbf{x}_{\text{img}}^2; ... \mathbf{x}_{\text{img}}^n] \in \mathbb{R}^{(n+1) \times d}$ where $\mathbf{x}_{\text{cls}}$ represents the class token embedding, $\mathbf{x}_{\text{img}}^i$ represents the embedding of the $i$-th image patch, $n$ is the length of the image token sequence, and $d$ is the dimensionality of the hidden state, the encoder first transforms $\mathbf{X}$ into queries ($\mathbf{Q} = \mathbf{X}\mathbf{W}_Q$), keys ($\mathbf{K} = \mathbf{X}\mathbf{W}_K$), and values ($\mathbf{V} = \mathbf{X}\mathbf{W}_V$) using three weight matrices $\mathbf{W}_Q, \mathbf{W}_K, \mathbf{W}_V \in \mathbb{R}^{d \times d}$. Subsequently, the attention matrix is computed as $\mathbf{A} = \text{softmax}\left(\mathbf{Q}\mathbf{K}^\top / \sqrt{d_k}\right)$ with output $\mathbf{O} = \mathbf{A}\mathbf{V}$. In this work, we refer to the first row of $\mathbf{A}$ as the [CLS] attention, representing the attention weights of the [CLS] token on all other tokens, which provides a measure of the importance of each visual token to the overall image representation.

## 3.2 MOTIVATION

### 3.2.1 THE CORE CHALLENGE: THE MISALIGNMENT OF IMPORTANCE SIGNALS

A fundamental challenge in visual token pruning lies in accurately defining and measuring the *"importance"* of a visual token. Existing methods predominantly rely on one of two signals:

1. ***Intra-modal Redundancy***: Discarding tokens that are highly similar to others, under the assumption that they contribute little new information.

2. ***Cross Attention***: Preserving tokens that receive high attention from text tokens within the Language Model, under the assumption that this indicates relevance to the user's instruction.

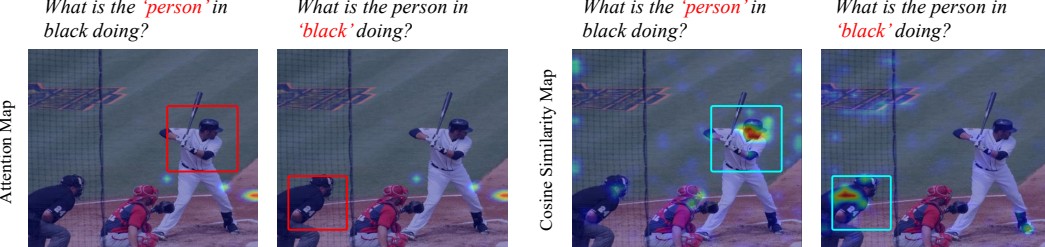

Figure 3: Comparison of attention and cosine similarity visualizations between key text and image.

However, we identify a critical misalignment in the latter signal. The text-visual attention mechanism within transformer-based LLMs is primarily designed for feature fusion, not for accurate spatial grounding. As a result, it often exhibits a positional bias, where attention scores do not reliably correlate with the semantic relevance between a specific text token and a specific image region. This phenomenon, which we refer to as text-visual attention shift, is visualized in Figure 3. The visual tokens with the highest attention scores frequently fail to correspond to the image regions containing the objects or attributes mentioned in the key instruction tokens.

### 3.2.2 RE-ESTABLISHING ALIGNMENT VIA SEMANTIC SIMILARITY

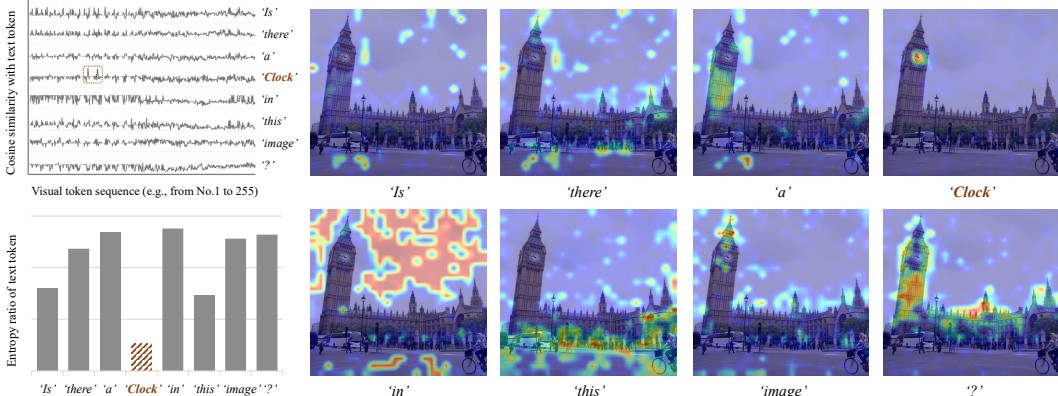

Figure 4: (Top-left) Line plots of cosine similarity for each text token against all visual tokens. (Bottom-left) The computed Entropy-Ratio ($ER$) value for each text token. Those with anomalously low $ER$ values are identified as key text tokens. (Right) Visualization of the cosine similarity map.

Based on the observation of attention shift, we employ cosine similarity as a more reliable measure for gauging cross-modal relevance. Furthermore, as shown in Figure 4 (Right),we find that not all text tokens contribute equally to locating key image regions. Common words like "there", "is", and "in" add noise and interfere with proper visual token selection. This necessitates a filtering process: *first identifying the most relevant text tokens from user instructions* to guide the visual pruning. Our analysis reveals that the similarity distribution between a key text token and all visual tokens exhibits a distinct, peaked outlier (Top-left of Figure 4), whereas the distributions for non-key tokens are relatively uniform. This contrast provides a clear signal for identifying vision-critical tokens.

To quantify this difference, we employ the entropy of the similarity distribution to measure the uncertainty of a text token's visual associations. To further accentuate the distinction, we introduce a novel *entropy-ratio* metric ($ER$), defined as the entropy divided by the ratio of its maximum to mean similarity ($\frac{max}{mean}$). This effectively amplifies the difference between focused, vision-critical tokens and dispersed, non-critical ones. As shown in Figure 4 (Bottom-left), key tokens emerge as clear outliers in this $ER$ space, transforming the problem into an outlier detection task.

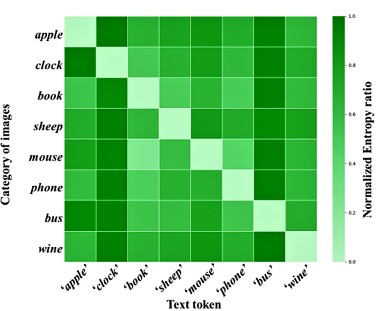

To further validate our hypothesis that the $ER$ metric effectively identifies vision-critical text tokens, we conducted a controlled experiment on the COCO 2014 dataset (Lin et al., 2014). We isolated images containing specific object classes and computed the $ER$ value for the corresponding class name text token against its associated images. As shown in Figure 5, for images that *contain* the specified object, the $ER$ value of the object's text token is consistently minimized. Conversely, for images that *do not contain* the object, the $ER$ value for the same text token is significantly larger. This empirical evidence strongly supports our core insight: the $ER$ metric serves as a robust and reliable indicator for identifying text tokens that are critically relevant to the visual content. A low $ER$ value signifies a strong and specific semantic alignment between a text token and the image, which is the fundamental basis of our token selection paradigm.

Figure 5: The entropy-ratio value between the object's text token and tokens of images containing different objects.

### 3.3 THE TEXTSCYTHE FRAMEWORK

Guided by the insights above, we propose **TextScythe**, a novel visual token pruning framework that first distills user instructions into critical text tokens and then uses them to select instruction-relevant visual tokens. Furthermore, to prevent excessive information loss, it complements this selection with diverse background tokens, ensuring both semantic relevance and contextual completeness. The overall architecture is illustrated in Figure 6. We now detail each component.

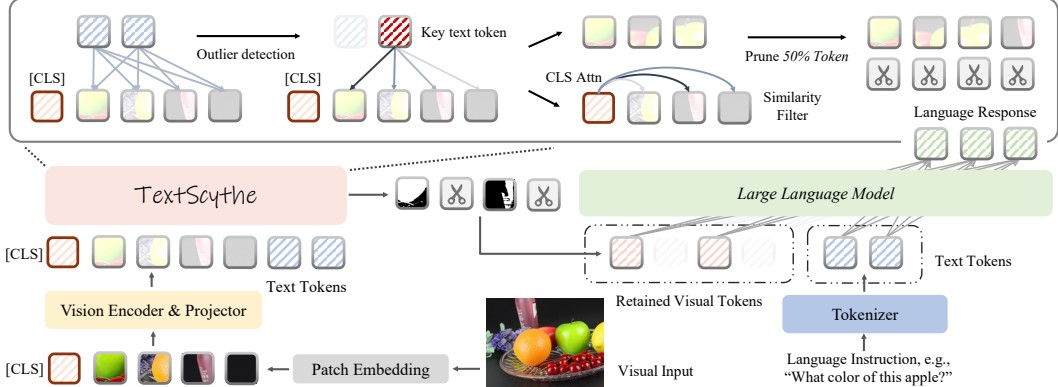

Figure 6: Framework of TextScythe. We first identify key text tokens using the entropy of cross-modal cosine similarity. Then, we select relevant visual tokens based on the similarity between key text tokens and visual tokens. Subsequently, we supplement the selected set with specific visual tokens having high attention scores to enhance the completeness of visual information.

#### 3.3.1 SELECTION OF TEXT-RELATED VISUAL TOKENS

To capture the relevance between user instructions and visual tokens, we first compute a cross-modal cosine similarity matrix $\mathbf{S}^{T2V} \in \mathbb{R}^{N_T \times N_V}$, where $N_T$ and $N_V$ denote the number of text and visual tokens respectively. The matrix is obtained through:

$$\mathbf{S}_{i,j}^{\mathrm{T2V}} = \frac{\mathbf{T}_i \cdot \mathbf{V}_j}{\|\mathbf{T}_i\| \cdot \|\mathbf{V}_j\|},\tag{1}$$

where $\mathbf{T}_i \in \mathbb{R}^D$ and $\mathbf{V}_j \in \mathbb{R}^D$ are the $D$-dimensional embedding vectors of the $i$-th text token and $j$-th visual token, respectively.

Following the insight that not all text tokens are equally important, we introduce an adaptive mechanism to identify the vision-critical text tokens. We first convert the similarity matrix into a probability distribution via row-wise softmax normalization:

$$\mathbf{P}_{ij} = \frac{\exp(\mathbf{S}_{ij}^{\text{T2V}})}{\sum_{k=1}^{N_V} \exp(\mathbf{S}_{ik}^{\text{T2V}})}. \tag{2}$$

For each text token $t_i$, we then calculate two key metrics: 1) the entropy of its similarity distribution $E_i = -\sum_j \mathbf{P}_{ij} \log \mathbf{P}_{ij}$, which measures the uncertainty of its visual associations; and 2) the ratio $R_i = \frac{\max_j \mathbf{P}_{ij}}{\text{mean}(\mathbf{P}_{i:})}$, which quantifies the peakedness of its similarity distribution. Then we calculated the entropy-ratio metric that combines these two measures:

$$ER_i = \frac{E_i}{R_i}. \tag{3}$$

Text tokens with exceptionally low $ER_i$ values indicate those with highly specific visual associations. We identify these vision-critical text tokens through adaptive thresholding:

$$\mathcal{T}_{\text{key}} = \{t_i \mid ER_i < \mu_{ER} - \lambda \cdot \sigma_{ER}\}, \tag{4}$$

where $\mu_{ER}$ and $\sigma_{ER}$ are the mean and standard deviation of $ER$ values across all text tokens, and $\lambda$ is a dynamic coefficient that adapts to instruction length:

$$\lambda = (\lfloor \log_{10}(N_{\text{inst}}) \rceil + 1) \times \alpha, \tag{5}$$

where $N_{\text{inst}}$ is the number of instruction tokens, $\lfloor \cdot \rceil$ denotes rounding to the nearest integer, and $\alpha$ is a scaling hyperparameter set to 0.7 based on empirical evaluation. This adaptive coefficient $\lambda$ intuitively tightens the selection criterion for longer instructions, which are more likely to contain non-visual, redundant words, thereby enhancing the robustness of key token identification.

For each key text token $t_i \in \mathcal{T}_{\text{key}}$, we then identify relevant visual tokens using adaptive thresholding on the probability distribution:

$$\mathcal{V}_{\text{rel}}^i = \{v_j \mid \mathbf{P}_{ij} > \mu_i + \beta \cdot \lambda \cdot \sigma_i\}, \tag{6}$$

where $\mu_i$ and $\sigma_i$ are the mean and standard deviation of $\mathbf{P}_{i:}$, and $\beta$ is a scaling hyperparameter that controls the overall sensitivity of visual token selection. This coordinated thresholding scheme is designed to enhance robustness against erroneous token selections. For longer instructions, although a larger $\lambda$ tightens the criterion for selecting text tokens, it remains possible that certain visually irrelevant text tokens are incorrectly included in $\mathcal{T}_{\text{key}}$. To mitigate the risk of such errors propagating to the visual token selection stage, we proportionally increase the visual threshold via the product $\beta \cdot \lambda$. This effectively raises the bar for visual tokens to be associated with any text token—including those potentially selected by mistake—thereby reducing the number of irrelevant visual tokens chosen and improving the overall stability of the pruning process across instructions of varying lengths.

The final set of instruction-relevant visual tokens is the union of these individual sets: $\mathcal{V}_{\text{rel}} = \bigcup_i \mathcal{V}_{\text{rel}}^i$. If the size of $\mathcal{V}_{\text{rel}}$ exceeds the target number of tokens to keep ($K$), we rank them by their aggregate similarity to all key text tokens ($\sum_{t_i \in \mathcal{T}_{\text{key}}} \mathbf{P}_{ij}$) and select the top-$K$ most relevant tokens.

### 3.3.2 SUPPLEMENTATION OF BACKGROUND VISUAL TOKENS

In addition to selecting instruction-relevant visual tokens ($\mathcal{V}_{\text{rel}}$), we supplement background tokens from the remaining visual tokens to prevent excessive information loss and maintain contextual completeness, thereby enhancing the model's robustness and scene understanding capability.

We leverage the attention mechanism from the visual encoder to assess the intrinsic importance of each visual token. For encoders with a [CLS] token (e.g., CLIP), we use the attention weights of the [CLS] token on all other tokens ($\mathbf{a}_v \in \mathbb{R}^{N_V}$). For encoders without a [CLS] token, we compute the mean attention across all tokens to obtain a similar importance measure.

Table 1: Performance comparison of various methods across different benchmarks. Results are shown for different pruning ratios, with accuracy and average performance highlighted. Best results in blue.

| Methods | GQA | MMB | MMB$_{CN}$ | MME | POPE | SQA | VQA$_{V2}$ | VQA$_{Text}$ | VizWiz | Average |
|---|---|---|---|---|---|---|---|---|---|---|
| Upper Bound, 576 Tokens | 61.9 | 64.7 | 58.1 | 1862 | 85.9 | 69.5 | 78.4 | 58.2 | 50.0 | 100% |
| LLaVA-1.5-7B | | | | *Budget = 192 Tokens; Token Pruning Rate = 66.7%* | | | | | | |
| ToMe (ICLR23) | 54.3 | 60.5 | - | 1563 | 72.4 | 65.2 | 68.0 | 52.1 | - | 88.5% |
| FastV (ECCV24) | 52.7 | 61.2 | 57.0 | 1612 | 64.8 | 67.3 | 67.1 | 52.5 | 50.8 | 90.5% |
| LLaVA-PruMerge (2024.5) | 54.3 | 59.6 | 52.9 | 1632 | 71.3 | 67.9 | 70.6 | 54.3 | 50.1 | 91.4% |
| PDrop (2024.10) | 57.1 | 63.2 | 56.8 | 1766 | 82.3 | 68.8 | 75.1 | 56.1 | 51.1 | 96.7% |
| FiCoCo-V (2024.11) | 58.5 | 62.3 | 55.3 | 1732 | 82.5 | 67.8 | 74.4 | 55.7 | 51.0 | 96.1% |
| MustDrop (2024.11) | 58.2 | 62.3 | 55.8 | 1787 | 82.6 | 69.2 | 76.0 | 56.5 | 51.4 | 97.2% |
| HiRED (AAAI25) | 58.7 | 62.8 | 54.7 | 1737 | 82.8 | 68.4 | 74.9 | 47.4 | 50.1 | 94.6% |
| SparseVLM (2025.2) | 57.6 | 62.5 | 53.7 | 1721 | 83.6 | 69.1 | 75.6 | 56.1 | 50.5 | 96.1% |
| DART (2025.2) | 58.9 | 63.6 | 57.0 | 1856 | 82.8 | 69.8 | 76.7 | 57.4 | 51.1 | 98.5% |
| TextScythe (Ours) | 60.0 | 63.1 | 57.3 | 1798 | 87.2 | 69.8 | 77.3 | 57.8 | 51.6 | 99.2% |
| LLaVA-1.5-7B | | | | *Budget = 128 Tokens; Token Pruning Rate = 77.8%* | | | | | | |
| ToMe (ICLR23) | 52.4 | 53.3 | - | 1343 | 62.8 | 59.6 | 63.0 | 49.1 | - | 80.4% |
| FastV (ECCV24) | 49.6 | 56.1 | 56.4 | 1490 | 59.6 | 60.2 | 61.8 | 50.6 | 51.3 | 85.4% |
| LLaVA-PruMerge (2024.5) | 53.3 | 58.1 | 51.7 | 1554 | 67.2 | 67.1 | 68.8 | 54.3 | 50.3 | 89.4% |
| PDrop (2024.10) | 56.0 | 61.1 | 56.6 | 1644 | 82.3 | 68.3 | 72.9 | 55.1 | 51.0 | 94.9% |
| FiCoCo-V (2024.11) | 57.6 | 61.1 | 54.3 | 1711 | 82.2 | 68.3 | 73.1 | 55.6 | 49.4 | 94.9% |
| MustDrop (2024.11) | 56.9 | 61.1 | 55.2 | 1745 | 78.7 | 68.5 | 74.6 | 56.3 | 52.1 | 95.7% |
| HiRED (AAAI25) | 57.2 | 61.5 | 53.6 | 1710 | 79.8 | 68.1 | 73.4 | 46.1 | 51.3 | 93.1% |
| SparseVLM (2025.2) | 56.0 | 60.0 | 51.1 | 1696 | 80.5 | 67.1 | 73.8 | 54.9 | 51.4 | 93.8% |
| DART (2025.2) | 57.9 | 63.2 | 57.0 | 1845 | 80.1 | 69.1 | 75.9 | 56.4 | 51.7 | 97.5% |
| TextScythe (Ours) | 59.1 | 62.5 | 56.9 | 1787 | 86.4 | 69.8 | 76.4 | 57.2 | 52.2 | 98.6% |
| LLaVA-1.5-7B | | | | *Budget = 64 Tokens; Token Pruning Rate = 88.9%* | | | | | | |
| ToMe (ICLR23) | 48.6 | 43.7 | - | 1138 | 52.5 | 50.0 | 57.1 | 45.3 | - | 70.1% |
| FastV (ECCV24) | 46.1 | 48.0 | 52.7 | 1256 | 48.0 | 51.1 | 55.0 | 47.8 | 50.8 | 76.7% |
| LLaVA-PruMerge (2024.5) | 51.9 | 55.3 | 49.1 | 1549 | 65.3 | 68.1 | 67.4 | 54.0 | 50.1 | 87.7% |
| PDrop (2024.10) | 41.9 | 33.3 | 50.5 | 1092 | 55.9 | 68.6 | 69.2 | 45.9 | 50.7 | 77.5% |
| FiCoCo-V (2024.11) | 52.4 | 60.3 | 53.0 | 1591 | 76.0 | 68.1 | 71.3 | 53.6 | 49.8 | 91.5% |
| MustDrop (2024.11) | 53.1 | 60.0 | 53.1 | 1612 | 68.0 | 63.4 | 69.3 | 54.2 | 51.2 | 90.1% |
| HiRED (AAAI25) | 54.6 | 60.2 | 51.4 | 1599 | 73.6 | 68.2 | 69.7 | 44.2 | 50.4 | 89.4% |
| SparseVLM (2025.2) | 52.7 | 56.2 | 46.1 | 1505 | 75.1 | 62.2 | 68.2 | 51.8 | 50.1 | 87.3% |
| DART (2025.2) | 55.9 | 60.6 | 53.2 | 1765 | 73.9 | 69.8 | 72.4 | 54.4 | 51.6 | 93.9% |
| TextScythe (Ours) | 56.5 | 61.2 | 55.7 | 1727 | 83.0 | 69.9 | 73.9 | 56.0 | 53.4 | 96.6% |

To ensure both importance and diversity in the supplemented tokens, we employ an iterative, similarity-inhibited selection process. We first sort the remaining visual tokens in descending order of their attention scores and select the token with the highest score into the supplementary set $\mathcal{V}_{sup}$.

For subsequent selections, we compute a comprehensive score $C_i$ for each candidate token that balances its own importance against its maximum similarity to any token already in the supplementary set, ensuring an optimal trade-off between information richness and diversity preservation:

$$C_i = \mathbf{a}_i - \max_{j \in \mathcal{V}_{sup}} \mathbf{S}_{ij}^{V2V}, \tag{7}$$

where $\mathbf{S}_{ij}^{V2V}$ is the cosine similarity between visual tokens $i$ and $j$, and $\mathbf{a}_i$ is the attention score for visual token $i$. The candidate with the highest comprehensive score $C_i$ is then added to $\mathcal{V}_{sup}$.

This process repeats iteratively until the total number of selected tokens ($|\mathcal{V}_{rel} \cup \mathcal{V}_{sup}|$) reaches the predefined target $K$. This strategy effectively maximizes the informational diversity of the final visual token set, ensuring comprehensive scene coverage while avoiding redundancy.

## 4 EXPERIMENTS

**Experiment Setting.** We conduct experiments on four MLLMs across nine image-based and three video-based benchmarks. For details on implementation, please refer to Appendix A.1.

### 4.1 MAIN RESULTS

We evaluate TextScythe on a wide range of vision-language benchmarks, including GQA, MMBench, MMBench-CN, MME, SQA, VQA$_{Text}$, VQA$_{V2}$, VizWiz, and hallucination-specific benchmarks

Table 2: Performance comparison of various methods across different benchmarks. Results are shown for different pruning ratios, with accuracy and average performance highlighted. Best results in blue.

| Methods | GQA | MMB | MMB$_{CN}$ | MME | POPE | SQA | VQA$_{V2}$ | VQA$_{Text}$ | VizWiz | Average |
|---|---|---|---|---|---|---|---|---|---|---|
| Upper Bound, 2880 Tokens | 64.2 | 67.4 | 60.6 | 1851 | 86.5 | 70.1 | 81.8 | 64.9 | 57.6 | 100% |
| LLaVA-NeXT-7B | | | | | *Retain 320 Tokens* (↓ **88.9%**) | | | | | |
| FastV (ECCV24) | 55.9 | 61.6 | 51.9 | 1661 | 71.7 | 62.8 | 71.9 | 55.7 | 53.1 | 88.0% |
| LLaVA-PruMerge (2024.5) | 53.6 | 61.3 | 55.3 | 1534 | 60.8 | 66.4 | 69.7 | 50.6 | 54.0 | 85.6% |
| PDrop (2024.10) | 56.4 | 63.4 | 56.2 | 1663 | 77.6 | 67.5 | 73.5 | 54.4 | 54.1 | 90.9% |
| MustDrop (2024.11) | 57.3 | 62.8 | 55.1 | 1641 | 82.1 | 68.0 | 73.7 | 59.9 | 54.0 | 92.2% |
| FasterVLM (2024.12) | 56.9 | 61.6 | 53.5 | 1701 | 83.6 | 66.5 | 74.0 | 56.5 | 52.6 | 91.1% |
| HiRED (AAAI25) | 59.3 | 64.2 | 55.9 | 1690 | 83.3 | 66.7 | 75.7 | 58.8 | 54.2 | 93.3% |
| SparseVLM (2025.2) | 56.1 | 60.6 | 54.5 | 1533 | 82.4 | 66.1 | 71.5 | 58.4 | 52.0 | 89.7% |
| GlobalCom$^2$ (2025.3) | 57.1 | 61.8 | 53.4 | 1698 | 83.8 | 67.4 | 76.7 | 57.2 | 54.6 | 92.2% |
| DART (EMNLP25) | 61.7 | 65.3 | 58.2 | 1710 | 84.1 | 68.4 | 79.1 | 58.7 | 56.1 | 93.9% |
| TextScythe (Ours) | 60.0 | 65.4 | 56.5 | 1771 | 86.8 | 71.6 | 77.2 | 59.2 | 54.8 | 95.8% |

**POPE and MME.** As shown in Tab. 1, TextScythe demonstrates strong robustness, consistently outperforming all competing methods including the recent DART. Notably, under an extreme pruning rate of 88.9% (keeping only 64 tokens), our method retains **96.6%** of the original performance, significantly surpassing DART (93.9%) and others. At intermediate pruning rates of 66.7% and 77.8%, TextScythe also achieves high retention rates of **99.2%** and **98.6%**, respectively.

Impressively, TextScythe performs especially well on hallucination benchmarks, scoring **83.0** on POPE at 88.9% pruning—nearly 10 points higher than the second-best method. This highlights its ability to preserve critical semantic information while aggressively removing redundant tokens. Moreover, on VizWiz and SQA, TextScythe even exceeds the unpruned baseline at various pruning ratios, indicating that it not only preserves but enhances model focus by filtering out visual noise.

## 4.2 TEXTSCYTHE WITH HIGHER RESOLUTION

For further comprehensive evaluation, we also assessed TextScythe on LLaVA-NeXT (Liu et al., 2024b) across all the aforementioned benchmarks, comparing it with current SOTA methods. LLaVA-NeXT introduces a high-resolution image processing strategy that generates substantially longer visual token sequences. To evaluate our method under this high-redundancy setting, we maintained a fixed budget of 320 visual tokens. As shown in Tab. 2, TextScythe achieves the top performance on multiple benchmarks and obtains the highest average performance retention of **95.8%**, substantially surpassing the current SOTA method DART (93.9%). These results confirm the superior capability and robustness of TextScythe in handling high-resolution visual inputs.

Table 3: Video QA Evaluations with 50% of visual tokens retained.

| Methods | TGIF-QA | | MSVD-QA | | MSRVT-QA | | Avgerge | |
|---|---|---|---|---|---|---|---|---|
| | Acc. | Score | Acc. | Score | Acc. | Score | Acc. | Score |
| LLaMA-Adapter 7B | - | - | 54.9 | 3.1 | 43.8 | 2.7 | - | - |
| VideoChat 7B | 34.4 | 2.3 | 56.3 | 2.8 | 45.0 | 2.5 | 45.1 | 2.5 |
| Video-LLaMA 7B | - | - | 51.6 | 2.5 | 29.6 | 1.8 | - | - |
| Video-ChatGPT 7B | 51.4 | 3.0 | 64.9 | 3.3 | 49.3 | 2.8 | 55.2 | 3.0 |
| Video-LLaVA 7B | 47.0 | 3.4 | 70.2 | 3.9 | 57.3 | 3.5 | 58.2 | 3.6 |
| + FastV | 45.2 | 3.1 | 71.0 | 3.9 | 55.0 | 3.5 | 57.1 | 3.5 |
| + DART | 46.3 | 3.3 | 71.0 | 4.0 | 56.7 | 3.6 | 58.0 | 3.7 |
| + TextScythe (Ours) | 46.2 | 3.4 | 70.8 | 3.9 | 57.1 | 3.5 | 58.0 | 3.6 |

Additionally, on video understanding benchmarks, TextScythe remains competitive with top methods like DART and surpasses other efficient approaches (Tab. 3). This further validates the applicability of our method in handling sequential and high-resolution visual inputs under constrained token budgets.

## 4.3 EFFICIENCY ANALYSIS

To assess the practical efficiency of TextScythe, we compare total inference time, prefill time, end-to-end latency, GPU memory usage, and accuracy on LLaVA-1.5-7B under a 90% pruning ratio. As shown in Figure 7, TextScythe achieves a significant 42.1% reduction in total inference time (from 49:41 to 28:42) and a 40.7% decrease in latency compared to the unpruned model, while maintaining 95.9% of the original accuracy. Compared to FastV, TextScythe not only runs faster but also uses less memory while delivering higher accuracy. These results highlight the practical strength of our method in achieving an optimal balance between accuracy and efficiency. The substantial speedup, coupled

with lower memory consumption, demonstrates that TextScythe is highly suitable for real-world deployment scenarios where both computational resources and model performance are critical.

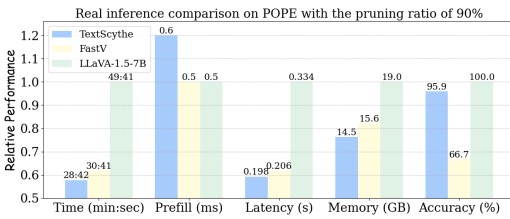

Figure 7: Efficiency Analysis.

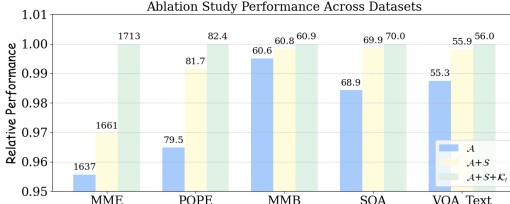

Figure 8: Impact of different components.

### 4.4 ABLATION STUDY

To validate the effectiveness of each component in our method, we conducted ablation experiments. All experiments were performed on the LLaVA-1.5-7B, with a visual token reduction rate of 90%. Specifically, we assess the following configurations: using only [CLS] attention ($\mathcal{A}$), incorporating similarity suppression between selected tokens ($\mathcal{A} + \mathcal{S}$), and the full combination with key text token guidance ($\mathcal{A} + \mathcal{S} + \mathcal{K}_t$). As shown in Figure 8, compared to using [CLS] attention alone ($\mathcal{A}$), integrating similarity suppression ($\mathcal{A} + \mathcal{S}$) improves performance across multiple benchmarks. The full model ($\mathcal{A} + \mathcal{S} + \mathcal{K}_t$) achieves the best results on all benchmarks, confirming that combining visual diversity through similarity suppression with textual relevance through key token guidance is crucial for effective pruning. In addition, we investigate the impact of key hyperparameters on model performance. As shown in Figure 9 and Figure 10, both excessively large and small values lead to suboptimal results. Detailed analysis and discussion can be found in Appendix A.2

### 4.5 TEXTSCYRHE WITH OTHER VLM ARCHITECTURE

To verify the architectural generalization of TextScythe beyond LLaVA-based models, we conduct experiments on the Qwen2.5-VL-7B (Bai et al., 2025a) architecture. As shown in Tab. 4, TextScythe demonstrates strong generalization capability across this architecture, consistently outperforming the text-visual attention-based FastV at various reduction ratios, highlighting its robustness and adaptability to different model designs. Notably, it achieves average performance retention rates of 97.6%, 94.0%, and 89.2% at 66.7%, 77.8%,

Table 4: Comparative Experiments on Qwen2.5-VL-7B.

| Methods | MMB | MME | POPE | SQA | VQA$_{\text{Text}}$ | Avg. |
|---|---|---|---|---|---|---|
| Upper Bound | 82.8 | 2304 | 86.1 | 84.7 | 84.8 | 100% |
| Qwen2.5-VL-7B | | | *Token Pruning Rate = 66.7%* | | | |
| FastV (ECCV24) | 75.7 | 2072 | 82.2 | 78.5 | 77.9 | 92.3% |
| TextScythe (Ours) | 81.4 | 2263 | 86.6 | 82.5 | 79.2 | 97.6% |
| Qwen2.5-VL-7B | | | *Token Pruning Rate = 77.8%* | | | |
| FastV (ECCV24) | 74.9 | 2036 | 80.7 | 78.0 | 69.0 | 89.2% |
| TextScythe (Ours) | 80.8 | 2177 | 85.5 | 81.2 | 70.1 | 94.0% |
| Qwen2.5-VL-7B | | | *Token Pruning Rate = 88.9%* | | | |
| FastV (ECCV24) | 69.2 | 1940 | 78.6 | 77.4 | 60.3 | 84.3% |
| TextScythe (Ours) | 76.2 | 2066 | 82.4 | 80.4 | 62.3 | 89.2% |

and 88.9% token reduction rates respectively, significantly higher than FastV's 92.3%, 89.2%, and 84.3%. These results prove that TextScythe's entropy-based pruning strategy effectively generalizes across different VLM architectures.

## 5 CONCLUSION

This work introduces TextScythe, a visual token pruning framework that tackles the core issue of attention shift, through estimating importance with cross-modal similarity. It identifies key text tokens, with an entropy-based metric, and uses their cosine similarity to visual tokens to guide pruning, coordinated by an adaptive threshold. This reframes pruning as semantic-aware content selection. Extensive experiments demonstrate that our proposed TextScythe maintains strong performance at high pruning rates, offering an efficient paradigm to accelerate LVLM inference.

*We illustrate the broader impact of TextScythe and LLM usage in Section B and D, respectively.*

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

## A APPENDIX

## ∞ Technical Appendices and Supplements

In this appendix, we provide detailed information regarding the experimental setup, encompassing the datasets, model architectures, and comparison methods. Then, we offer a detailed analysis and discussion of the impact of hyperparameters on model performance.

### A.1 DETAILED EXPERIMENT SETTINGS

#### A.1.1 DATASETS

We conducted experiments on several widely used visual understanding benchmarks. For image understanding task, we performed experiments on ten widely used benchmarks, including GQA (Hudson & Manning, 2019), MMBench (MMB) and MMB-CN (Liu et al., 2025b), MME (Fu et al., 2023), POPE (Li et al., 2023), VizWiz (Bigham et al., 2010), SQA (ScienceQA) (Lu et al., 2022), $VQA_{V2}$ (VQA V2) (Goyal et al., 2017) and $VQA_{Text}$ (TextVQA) (Singh et al., 2019)

**GQA.** (Hudson & Manning, 2019) The GQA benchmark is composed of three parts: scene graphs, questions, and images. The image part contains images, as well as the spatial features of images and the features of all objects in images. The questions in GQA are designed to test the understanding of visual scenes and the ability to reason about different aspects of an image.

**MMBench.** (Liu et al., 2025b) The MMBench benchmark comprehensively evaluates the model's overall performance across multiple dimensions. It includes three levels of ability dimensions. The first level (L-1) consists of two main abilities, perception and reasoning. The second level (L-2) expands based on the first level, including six sub-abilities. The third level (L-3) further refines the second level, encompassing 20 specific ability dimensions. This hierarchical structure enables a granular and comprehensive evaluation of the model's various capabilities.

**MME.** (Fu et al., 2023) The MME benchmark is also a comprehensive benchmark meticulously designed to thoroughly evaluate various aspects of a model's performance. It consists of 14 subtasks that specifically aim to evaluate both the model's perceptual and cognitive abilities. By utilizing manually constructed instruction-answer pairs and concise instruction design, it effectively mitigates issues such as data leakage and unfair evaluation of model performance.

**POPE.** (Li et al., 2023) The POPE benchmark is primarily used to evaluate the degree of Object Hallucination in models. It reformulates hallucination evaluation by requiring the model to answer a series of specific binary questions regarding the presence of objects in images. Accuracy, Recall, Precision, and F1 Score are effectively employed as reliable evaluation metrics to precisely measure the model's hallucination level under three different sampling strategies.

**ScienceQA.** (Lu et al., 2022) The ScienceQA benchmark covers a rich diversity of domains, including natural science, language science, and social science. Within each subject, questions are categorized first by the topic, then by the category, and finally by the skill. This hierarchical categorization results in 26 topics, 127 categories, and 379 skills, providing a comprehensive and diverse range of scientific questions. It provides a comprehensive evaluation of a model's capabilities in multimodal understanding, multi-step reasoning, and interpretability.

**VQA-v2.** (Goyal et al., 2017) The VQA-v2 benchmark evaluates the model's visual perception capabilities through open-ended questions. It consists of 265,016 images, covering a wide variety of real-world scenes and objects, providing rich visual contexts for the questions. For each question, there are 10 ground truth answers provided by human annotators, which allows for a comprehensive evaluation of the performance of different models in answering the questions accurately.

**TextVQA.** (Singh et al., 2019) The TextVQA benchmark focuses on the comprehensive integration of diverse text information within images. It meticulously evaluates the model's text understanding and reasoning abilities through a series of visual question-answering tasks with rich textual information. Models need to not only understand the visual content of the images but also be able to read and reason about the text within the images to answer the questions accurately.

### A.1.2 MODELS

We evaluate TextScythe using various open-source MLLMs. For image understanding tasks, experiments are conducted on the LLaVA family, including LLaVA-1.5-7B[1] (Liu et al., 2024a) and LLaVA-Next-7B[2] (Liu et al., 2024b), with the latter used to validate performance on high-resolution images. Furthermore, we validate our method on other advanced model Qwen2.5-VL-7B (Bai et al., 2025a). For video understanding tasks, we use Video-LLaVA (Lin et al., 2023) as the baseline model. following the settings reported in their paper to ensure a fair comparison.

### A.1.3 BASELINES

We analyze multiple representative methods for accelerating multi-modal language models (MLLMs) through token reduction. These methods share the goal of improving efficiency by reducing redundant tokens, yet differ in their strategies, such as token merging, pruning, or adaptive allocation.

**ToMe** (Bolya et al., 2022) merges similar tokens in visual transformer layers through lightweight matching techniques, achieving acceleration without requiring additional training.

**FastV** (Chen et al., 2024) focuses on early-stage token pruning by leveraging attention maps, effectively reducing computational overhead in the initial layers.

**SparseVLM** (Zhang et al., 2024c) ranks token importance using cross-modal attention and introduces adaptive sparsity ratios, complemented by a novel token recycling mechanism.

**HiRED** (Arif et al., 2024) allocates token budgets across image partitions based on CLS token attention, followed by the selection of the most informative tokens within each partition, ensuring spatially aware token reduction.

**LLaVA-PruMerge** (Shang et al., 2024) combines pruning and merging strategies by dynamically removing less important tokens using sparse CLS-visual attention and clustering retained tokens based on key similarity.

**PDrop** (Xing et al., 2024) adopts a progressive token-dropping strategy across model stages, forming a pyramid-like token structure that balances efficiency and performance.

**FasterVLM** (Zhang et al., 2024b) evaluates token importance via CLS attention in the encoder and performs pruning before interaction with the language model, streamlining the overall process.

**MustDrop** (Liu et al., 2024d) integrates multiple strategies, including spatial merging, text-guided pruning, and output-aware cache policies, to reduce tokens across various stages.

**GlobalCom$^2$** (Liu et al., 2025a) introduces a hierarchical approach by coordinating thumbnail tokens to allocate retention ratios for high-resolution crops while preserving local details.

**DART** (Wen et al., 2025b) introduces a duplication-aware token reduction method that selects a small subset of pivot tokens, calculates cosine similarity between pivot tokens and remaining tokens, retains those with the lowest duplication to pivots, achieving significant acceleration while maintaining performance and good compatibility with efficient attention operators.

These methods collectively highlight diverse approaches to token reduction, ranging from attention-based pruning to adaptive merging, offering complementary solutions for accelerating MLLMs.

---

[1] https://huggingface.co/liuhaotian/llava-v1.5-7b
[2] https://huggingface.co/liuhaotian/llava-v1.6-vicuna-7b

### A.1.4 IMPLEMENTATION DETAILS

All of our experiments are conducted on Nvidia A800-80G GPU. The implementation was carried out in Python 3.10, utilizing PyTorch 2.1.2, and CUDA 11.8. All baseline settings follow the original paper. Our hyperparameter designs are $\alpha$=0.7 and $\beta$=2.0, respectively.

### A.2 IMPACT OF HYPERPARAMETE

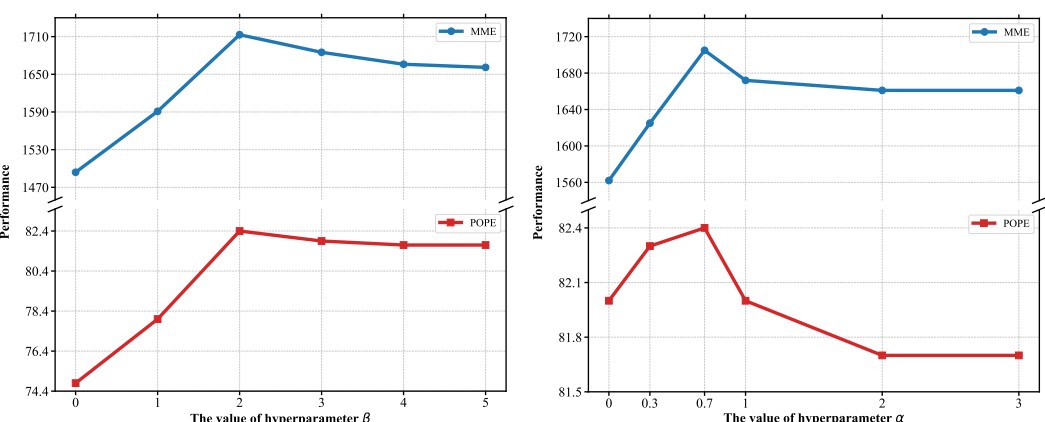

Figure 9: Impact of hyperparamete $\beta$.  Figure 10: Impact of hyperparamete $\alpha$.

We compared the impact of different hyperparameters on model performance. First, with the text token selection parameter $\alpha$ fixed at 0.7, we varied the visual token selection parameter $\beta$ (Figure 9). For both MME and POPE, performance improved significantly as $\beta$ increased from 0 to 2, indicating that a moderate $\beta$ value effectively filters out irrelevant visual tokens while retaining critical ones. However, when $\beta$ exceeded 2, performance gradually declined, suggesting that overly strict visual token selection may remove important information, leading to suboptimal results. This highlights the importance of balancing visual token selectivity to avoid both noise retention and over-pruning.

Subsequently, we fixed the visual token selection threshold $\beta \cdot \lambda$ at 2.4 to isolate the effect of the text token selection parameter $\alpha$ (Figure 10). For both MME and POPE, performance improved as $\alpha$ increased from 0 to 0.7, reaching an optimum. We hypothesize that when $\alpha$ is too small, the text token selection becomes overly lenient, allowing non-visual or irrelevant text tokens (e.g., functional words) to be incorrectly identified as key tokens, which in turn guides the visual token selection toward erroneous regions. Conversely, when $\alpha$ exceeds 0.7, the text token selection becomes overly strict, potentially excluding legitimate key text tokens that are essential for capturing instruction-relevant visual content. This leads to a gradual performance decline, as critical visual information may be omitted. The peak at $\alpha = 0.7$ demonstrates that our adaptive thresholding mechanism effectively balances the trade-off between inclusivity and precision in text token selection, ensuring that only the most vision-critical tokens are used to guide pruning. This result underscores the importance of properly calibrating $\alpha$ to maximize the synergy between text and visual token selection.

In summary, both hyperparameters require careful tuning to achieve optimal performance. The trends confirm that TextScythe's effectiveness relies on a coordinated balance between text token selection (controlled by $\alpha$) and visual token selection (controlled by $\beta$). While the method shows robustness through adaptive design, the optimal values underscore the importance of appropriate thresholds for minimizing errors in token selection.

## B ETHICS STATEMENT

This work presents a method for improving the computational efficiency of vision-language models through token pruning. We recognize the following ethical considerations:

**Positive Impacts:** Our method can reduce the computational cost and energy consumption of large AI models, contributing to more environmentally sustainable AI deployment. This could make advanced AI capabilities more accessible in resource-constrained environments.

**Potential Concerns:** While token pruning generally preserves model performance, aggressive pruning might potentially amplify biases or affect model fairness by disproportionately removing information about underrepresented visual concepts. However, our experiments show that TextScythe maintains robust performance across diverse benchmarks.

**Data Usage:** Our research uses publicly available benchmarks and models. All datasets employed in this study are widely used in the research community for non-commercial purposes.

**Broader Implications:** We believe the efficiency improvements offered by our method align with responsible AI development goals by reducing the computational barrier to using advanced multimodal AI systems

## C    REPRODUCIBILITY STATEMENT

To ensure the reproducibility of our work, we provide the following:

**Code Availability:** The implementation of TextScythe will be made publicly available upon publication.

**Experimental Details:**

- Complete hyperparameter settings for all experiments are provided in Appendix A.1.4.

- The detailed method implementation process is described in Section 3.3.

- The specific versions of all baseline methods we compared against are clearly cited.

**Datasets:** All datasets used in this study are publicly available.

**Models:** Our experiments use publicly available model checkpoints.

**Computational Resources:** We report the specific hardware configurations and computational requirements in Appendix A.1.4. All experiments can be reproduced with similar GPU resources.

## D    THE USE OF LARGE LANGUAGE MODELS (LLMS)

In preparing this manuscript, we utilized DeepSeek-R1 as a writing and editing assistant. Its role was limited to enhancing the clarity and fluency of the English in various sections. All scientific ideas, research methodology, experimental design, result analysis, and technical contributions are solely the product of the human authors. DeepSeek was not involved in any aspect of research conception, algorithm design, data interpretation, or validation of mathematical formulations, theoretical analyses, and experimental results.

