# OpenReview forum: "Spot the Critical Words: Text-Guided Visual Token Pruning for Efficient Large Vision-Language Model Inference"
_ICLR.cc/2026/Conference — ICLR 2026 Conference Withdrawn Submission_

### Official Review · Reviewer_sSUb · 2025-10-28

**Soundness:** 2
**Presentation:** 2
**Contribution:** 2
**Rating:** 4
**Confidence:** 4

**Summary:**

Large Vision-Language Models (LVLMs) achieve remarkable performance in multimodal tasks but suffer from high computational overhead due to massive visual tokens.  To address these issues, the paper proposes TextScythe, a plug-and-play visual token pruning framework that optimizes both instruction relevance and feature diversity. It first identifies vision-critical text tokens via entropy-based cross-modal similarity analysis, selects instruction-relevant visual tokens, and supplements background tokens to preserve contextual completeness.
TextScythe achieves state-of-the-art performance across 9 image benchmarks (e.g., GQA, POPE) and 3 video benchmarks (e.g., MSVD-QA).

**Strengths:**

1. A major strength lies in the simplicity and intuitiveness of TextScythe’s core idea, without compromising originality. Instead of overcomplicating pruning with opaque mathematical models, it anchors on a relatable intuition: prioritize visual tokens tied to "vision-critical" text (via the entropy-ratio metric) and supplement necessary background.
2. The paper exhibits good clarity in structure, which strengthens its readability and rigor.
3. The TextScythe delivers compelling SOTA results across 9 image benchmarks and 3 video benchmarks.

**Weaknesses:**

1. Insufficient depth in analyzing limitations of existing methods and inadequate argumentation. The paper classifies existing pruning methods into "attention-based" and "similarity-based" paradigms and briefly mentions their "limitations" (e.g., attention shift, ignorance of cross-modal interaction), but lacks in-depth analysis of the root causes of these problems. For instance, when claiming that "text-visual attention in transformers is designed for feature fusion rather than spatial grounding," the authors fail to provide supporting evidence, and do not cite relevant studies nor conduct auxiliary experiments (e.g., visualizing attention maps vs. actual instruction-relevant regions) to verify the "attention shift" phenomenon.

2. Limited Innovation. TextScythe is positioned as an "instruction-aware" pruning method, but its core logic still revolves around similarity calculations, whether it is the text-visual cosine similarity for key token selection or visual-visual similarity for background supplementation. This does not fundamentally deviate from the "similarity-based" paradigm it critiques. Moreover, the proposed entropy-ratio (ER) metric, though labeled "novel," is essentially a statistical refinement of similarity distributions and is heavily tied to object-level semantic relevance (e.g., ER targets text tokens describing objects, such as "cat" or "car"). For abstract instructions (e.g., "describe the atmosphere of the scene"), the ER metric may fail to identify meaningful text tokens, as abstract semantics lack clear visual similarity anchors.

3. The paper has two critical gaps in experiments:
- First, while it mentions compatibility with "high-resolution inputs" (e.g., LLaVA-NeXT), it only reports results for a single high-resolution setting (fixed input size) and lacks cross-resolution comparative experiments. For example, it does not test how TextScythe performs when input resolution varies (e.g., 256×256, 512×512, 1024×1024). Without this data, it is unclear whether the method’s pruning logic (e.g., ER threshold, background supplementation ratio) is robust to resolution changes.
- Second, the framework involves multiple hyperparameters but provides no hyperparameter analysis. The authors do not explain how the rule is determined, nor do they conduct ablation experiments (e.g., testing $\lambda = 0.5, 1.0, 1.5$) to show that the chosen parameters are optimal. This makes the method’s reproducibility questionable, as other researchers may struggle to tune parameters for different models/datasets.

**Questions:**

Please see weaknesses. I will appreciate if the authors explain or improve these points in the rebuttal.

---

### Official Review · Reviewer_sL8L · 2025-10-29

**Soundness:** 2
**Presentation:** 3
**Contribution:** 2
**Rating:** 2
**Confidence:** 4

**Summary:**

This paper proposes the TextScythe method to prune visual tokens, thereby accelerating inference for large vision-language models.

Specifically, it first identifies the most important text tokens and then prunes the least relevant visual tokens with these text tokens.
Moreover, it supplements background tokens based on attention scores.

Experiments on various benchmarks, like GQA, MMB, MME, POPE, SQA, and VQA, are conducted, showing improved efficiency.

**Strengths:**

(1) This paper writes clearly and is easy to follow.
(2) The method is evaluated on 9 benchmarks and several large vision-language models.

**Weaknesses:**

(1) Pruning visual tokens with critical text token guidance is the main contribution of this work. However, as shown in Figure 8, the text token guidance has little effect on model performance (improvements from 60.8 to 60.9 on MMB, from 69.9 to 70.0 on SQA, from 55.9 to 56.0 on VAQ_Text).
It appears that pruning visual tokens based on similarity suppression can already achieve a suitable trade-off between performance and efficiency.

(2) Figure 3 shows visual comparisons between attention and cosine similarity.
    How to derive the attention and cosine similarity scores in the Figure?
    It is confusing because attention scores are also softmax-normalized cosine similarity scores.

(3) The paper proposes the entropy-ratio (ER) metric to evaluate the importance of text tokens. However, its design seems a little bit arbitrary. The entropy could already quantify the peakedness of the similarity distribution. Why is it necessary to be divided by the mean score?

(4) Table 1 uses the token pruning ratio to compare efficiency. However, pruning tokens in different layers could have different effects on efficiency. It would be better to compare efficiency in terms of inference time or FLOPs.

**Questions:**

See weaknesses.

---

### Official Review · Reviewer_C6Vp · 2025-10-31

**Soundness:** 2
**Presentation:** 2
**Contribution:** 2
**Rating:** 4
**Confidence:** 4

**Summary:**

This paper introduces TextScythe, a text-guided visual token pruning approach for accelerating LVLM inference. TextScythe first identifies vision-critical text tokens, then selects text-related visual tokens based on outlier-level similarity. Finally, it supplements a number of background tokens to preserve contextual completeness. Experimental results on various benchmarks validate the effectiveness of the proposed approach.

**Strengths:**

1. The experimental results validate the effectiveness of the proposed approach.
2. The paper is generally well-written and easy to read.

**Weaknesses:**

1. The proposed approach appears to mainly combine existing methods rather than introducing fundamentally new ideas. For example, the selection of text-related visual tokens resembles the strategies used in SparseVLM [1] and FastV [2]. A more detailed discussion on the technical novelty and distinctions from these works would strengthen the paper.
2. No quantitative experiments are provided to validate the text–visual attention shift phenomenon. The authors present only a single illustrative example, whereas a more systematic analysis across different models and datasets is needed to substantiate this claim.
3. How does the proposed approach perform on referring grounding tasks (e.g., RefCOCO benchmarks)? To further validate its effectiveness, results on fine-grained datasets such as DocVQA and InfoVQA would also be valuable.
4. More baselines should be included for comparison in the Qwen2.5-VL experiments.
5. Does the proposed approach also accelerate training while maintaining performance?

[1] SparseVLM: Visual Token Sparsification for Efficient Vision-Language Model Inference. https://arxiv.org/abs/2410.04417.

[2] An Image is Worth 1/2 Tokens After Layer 2: Plug-and-Play Inference Acceleration for Large Vision-Language Models. https://arxiv.org/abs/2403.06764.

**Questions:**

See the weaknesses above.

---

### Note · Authors · 2025-12-24

I have read and agree with the venue's withdrawal policy on behalf of myself and my co-authors.